# Safety Assessment of Stem Cell-Based Therapies: Current Standards and Advancing Frameworks

**DOI:** 10.3390/cells14211660

**Published:** 2025-10-23

**Authors:** Georgy E. Leonov, Lydia R. Grinchevskaya, Oleg V. Makhnach, Marina V. Samburova, Dmitry V. Goldshtein, Diana I. Salikhova

**Affiliations:** 1Federal Research Center of Nutrition, Biotechnology and Food Safety, 109240 Moscow, Russia; 2Research Institute of Molecular and Cellular Medicine of the Medical Institute Peoples’ Friendship, University of Russia, 117198 Moscow, Russiamarina_samburova33@mail.ru (M.V.S.); 3Research Centre for Medical Genetics, 115478 Moscow, Russia; buben6@yandex.ru (O.V.M.); dvgoldshtein@gmail.com (D.V.G.)

**Keywords:** regenerative medicine, cell therapy, biosafety, toxicity, biodistribution, immunogenicity, tumorigenicity, oncogenicity, cell product quality

## Abstract

Regenerative medicine is a rapidly evolving field of contemporary biomedical research that offers new therapeutic strategies for conditions previously considered untreatable. Cell therapy shows particular potential in this domain. However, rigorous biosafety measures are required in its development and clinical application. This review proposes a practice-oriented biosafety framework for cell therapy, translating key risks into operational principles: toxicity, oncogenicity/tumorigenicity/teratogenicity, immunogenicity, biodistribution; and cell product quality. For each principle, preclinical approaches and regulatory expectations are summarized. Criteria for immunological safety are addressed, including activation of innate immunity (complement, T- and NK-cell responses) and the need for HLA typing. Biodistribution assessment involves the use of quantitative PCR and imaging techniques (PET, MRI) to monitor cell fate over time. The risks of oncogenicity, tumorigenicity, and teratogenicity can be analyzed using a combination of in vitro methods and in vivo models in immunocompromised animals. Product quality assessment includes sterility, identity, potency, viability, and genetic stability, with alignment of procedures to regulatory requirements and an emphasis on quality-by-design principles to ensure safe and reproducible clinical use. Integrating toxicity and safety pharmacology data supports a balanced risk–benefit assessment and clinical trial planning.

## 1. Introduction

Over the past decade, regenerative medicine has emerged as a global scientific priority, with a growing emphasis on novel methods of treating various medical conditions. One of the promising areas of regenerative medicine is cell therapy [1]. Cell therapy is defined as the transfer of autologous or allogeneic cellular material into a patient for the purpose of medical intervention [2]. Currently, there are two main directions of cell therapy. Firstly, the utilization of cells or tissue-engineered constructs to restore damaged organs and tissues; tissue-specific (fibroblasts, chondrocytes, melanocytes) [3,4,5] and stem cells (including hematopoietic stem cells (HSCs)) [6], mesenchymal stem/stromal cells (MSCs) [7], epithelial stem cells from intestinal crypts [8], retinal stem cells [9], induced pluripotent stem cells (iPSCs), and embryonic stem cells [10]) are used for these purposes. Secondly, the administration of modified immune cells (Chimeric Antigen Receptor T-Cell (CAR-T), Natural killer (NK), and others) for the treatment of certain types of cancer [11,12].

The vast majority of Food and Drug Administration (FDA)-approved cell therapy approaches involve the use of immune cell preparations. Afamitresgene autoleucel (Tecelra, United States, approved in August 2024), an engineered T-cell receptor therapy containing autologous T-cells modified to target the MAGE-A4 antigen, is approved by the FDA for treating adults with unresectable or metastatic synovial sarcoma, a type of solid tumor cancer, in patients who have received prior chemotherapy, meet specific human leukocyte antigens (HLAs) and antigen criteria [13]. In addition, Lifileucel (Amtagvi, United States, approved in February 2024), a tumor-infiltrating lymphocyte therapy containing autologous T-cells harvested from the patient’s own tumor and expanded ex vivo, is approved by the FDA for treating adults with unresectable or metastatic melanoma, another solid tumor cancer, in those previously treated with PD-1 inhibitors and, if applicable, BRAF/MEK inhibitors [14]. Another cell-based drug that has received FDA authorization is the allogeneic bone marrow-derived MSCs product Rexlemestrocel-L, which is labeled for the treatment of steroid-refractory acute graft-versus-host disease (GVHD) in pediatric patients [15]. Prominent examples of successful applications of stem cells include the use of MSCs in fibrin gel for the treatment of inflammatory bowel disease (IBD) and Crohn’s disease. Researchers have demonstrated that the combination of MSCs and fibrin gel can assist in closing fistulas in patients with IBD [16]. Additionally, the use of pluripotent stem cells has shown promise in diabetes treatment, where differentiated cells are used to create pancreatic endoderm cells as a basis for treating type I diabetes [17,18].

Despite the remarkable therapeutic and regenerative potential of cell therapy, the biosafety aspects of their clinical application remain an underexplored area requiring further investigation. To ensure the successful translation of biomedical cell products into clinical practice, a comprehensive evaluation of multiple critical parameters is essential [19]. For example, allogeneic HSCs (allo-HSC) transplantation has allowed researchers to study many aspects of the safety of cellular products in detail [20]. The transplantation process and the subsequent post-transplant period are associated with a high risk of complications, including toxic effects of conditioning, infections, and GVHD, which are the main causes of mortality associated with allo-HSCs transplantation [21]. The first attempts at transplantation resulted in an 80% mortality rate [22]. However, due to the benefits of the Center for International Blood and Marrow Transplant Research (CIBMTR), mortality during transplantation has been reduced. These achievements include the introduction of low-intensity conditioning regimens, the use of HLA-matched donors as a transplant source, improvements in concomitant therapy, and a reduced likelihood of developing acute GVHD [23].

A thorough biosafety assessment must include an analysis of biodistribution patterns, which involves tracking the movement and distribution of cells within the recipient. The evaluation of toxicity profiles is essential and should include both general systemic and local adverse effects that may occur following cell administration. Monitoring proliferative activity is crucial to understanding how cells multiply and behave after transplantation [24,25]. The oncogenic potential of cell products requires careful assessment to evaluate the risk of malignant transformation, while teratogenic effects must be considered, particularly when working with pluripotent cells that have the capacity to differentiate into various tissue types [26]. Immunogenicity studies are essential to understand how transplanted cells interact with the recipient’s immune system, and cell survival rates must be measured to determine post-implantation viability in target tissues or organs [27]. Equally important is the rigorous confirmation of cellular product quality, which involves verifying that the cells are sterile, free from pyrogenic and infectious agents, and authentic. The authenticity of the cell population must be verified, and their functional activity must meet established criteria to ensure therapeutic efficacy [28].

This review aims to provide a comprehensive analysis of critical biosafety issues, identify prevailing challenges in the field, and evaluate the methodologies currently used to assess them.

## 2. Assessment of the Toxicity of the Cellular Product

The use of cells as therapeutic agents differs in several essential ways from drugs based on chemically synthesized or natural compounds. Cells under normal conditions cause virtually no direct cytotoxic effects, but have the potential to mediate tissue damage through various mechanisms [29]. The adverse effects associated with cell therapy can be broadly categorized into various key mechanisms, including immunological responses, tumorigenesis, cellular senescence, and administration-related complications. The mechanisms of action that lead to these adverse events are not yet fully understood [30].

The concept of cellular product toxicity refers to the degree of harmful effects that the cells and their components have on the recipient. This comprehensive assessment is essential for ensuring the safe translation of cell-based therapies from preclinical research to clinical application [31]. Potential side effects of cell therapy depend on many factors, such as the area of injection, the number and type of cells, and the patient’s condition. For example, it has been shown that after performing autologous stem cell transplantation into the vitreous body for the treatment of age-related macular degeneration, patients experienced severe vision loss, ocular hypertension, hemorrhagic retinopathy, and other complications. This type of treatment and protocol for performing procedures have not been approved, but this example highlights the need for a comprehensive assessment of the biosafety of cell therapy [32]. Current European Medicines Agency (EMA) guidelines on cell-based medicinal products stipulate that new cell-based products should be subject to general toxicity and biosafety pharmacology studies [33].

### 2.1. General and Reproductive Toxicity

Toxicity studies are primarily aimed at determining the relationship between drug exposure and adverse effects, usually taking structural tissue changes at post-mortem examination as the primary endpoint. Accordingly, a key goal of toxicity studies is to determine the maximum tolerated dose of a drug for single and repeated administration. For preclinical toxicity assessment, it is advisable to study both acute and chronic toxicity. The evaluation process involves multiple layers of investigation, starting with in vivo studies that require careful monitoring of various physiological parameters. Special attention should be given to evaluating mortality rates, as they provide critical insights into the acute and chronic effects of cell transplantation [34]. In addition to behavioral and physiological examinations, comprehensive laboratory testing is essential. Blood and urine tests play a pivotal role in toxicity evaluation, providing valuable information about organ function and systemic responses. Key blood parameters that are routinely monitored include a complete blood count with differential, biochemical parameters: albumin, aspartate aminotransferase, alanine aminotransferase, alkaline phosphatase levels indicating liver metabolism, blood urea nitrogen and creatinine assessing kidney function, electrolyte balance measuring calcium, sodium, potassium and phosphate levels, metabolic markers including lipid profile (cholesterol, lipoproteins, triglycerides) and glucose [35,36].

The pathophysiological analysis forms another critical component of toxicity analysis. This involves both macroscopic and microscopic examination of tissues to identify any structural or functional abnormalities. Multi-organ toxicity assessment includes histopathological examination of all major organ systems, with particular attention paid to organs in which cellular accumulation of the drug is observed based on biodistribution studies. Standardized toxicity scoring systems ensure that adverse events are consistently evaluated across studies [37]. Histological examination of tissue samples at the transplantation site is necessary to assess cell death, immune cell infiltration, and other pathological signs. It is also useful to perform a histological examination of the liver, lungs and kidneys, regardless of the location of the transplanted cells. The choice of animal models and experimental design should reflect the intended clinical application of the cellular product to ensure relevance and reliability of the results [38,39]. The assessment of immunotoxicity includes the evaluation of both intended and unintended effects on immune system function, including cytokine profiles, lymphocyte subset analysis, and functional immune tests. This analysis is particularly important for cellular products with immunomodulatory properties [40].

The overall clinical condition of the animals must be meticulously documented, including detailed observations of weight changes, behavioral patterns, and appetite. These parameters serve as early indicators of potential adverse reactions to the cellular product [41]. All analytical methods employed in cell therapy biosafety assessment must undergo rigorous validation according to International Conference on Harmonisation (ICH) guidelines. Validation parameters include accuracy, precision, linearity, range, specificity, and robustness. For cell-specific assays, additional considerations include matrix effects from biological specimens and stability of cellular analytes under various storage and processing conditions [42]. Thus, in the study of a cell preparation containing human regulatory macrophages (Mreg_UKR), detailed clinical, pharmacokinetic, pharmacological and toxicological tests were carried out on mice with immunodeficiency (NMRI-nude), during which the biosafety of Mreg_UKR therapy was demonstrated. Simultaneously, the authors underscore the heightened significance of clinical studies in evaluating the biosafety of cell therapy [43]. Assessment of neurological toxicity requires specific protocols, especially for cellular products administered intrathecally or intracerebrally. Neurological examination, histopathological examination of brain and spinal cord tissue, and examination of cerebrospinal fluid parameters provide a comprehensive analysis of neurological biosafety [44]. In a clinical trial of immunoablative autologous hematopoietic stem cell transplantation (IAHSCT) in multiple sclerosis, evidence of transient central nervous system (CNS) toxicity was found immediately following IAHSCT [45].

An assessment of reproductive toxicity may be warranted for some cell therapy products, particularly those with the potential to spread to germ cells or to exert hormonal effects. Histopathological examination of reproductive organs and analysis of reproductive parameters provide a comprehensive biosafety profile [44,46]. These studies evaluate both male and female fertility and developmental effects from conception through adulthood, using various guidelines such as the Organization for Economic Co-operation and Development’s (OECD’s) Extended One-Generation Reproductive Toxicity Study (EOGRTS) [47]. In a study of reproductive biosafety of therapy with human umbilical cord mesenchymal stem cells (UC-MSCs) in Sprague-Dawley rats. Assessments made included mortality, clinical observations, body weight, food consumption, fertility parameters of male and female, litter, and fetus parameters. It was shown that very high doses of the drug led to a number of undesirable effects, while no negative impact on the offspring was shown [48].

### 2.2. Safety Pharmacology

In contrast, biosafety pharmacology studies aim to predict the likelihood that a drug will be unsafe when administered to patients at therapeutic doses and thus aim to prevent such events. As part of this task, biosafety pharmacology studies aim to predict the possible occurrence of rare adverse effects [49].

Safety pharmacology panels for cardiovascular, respiratory, and neurologic systems play a crucial role in assessing drug safety during the development process [50]. The cardiovascular safety panel includes core measurements of hemodynamic parameters, including blood pressure, heart rate variability, electrocardiography (ECG) analysis, and cardiac contractility. Triggers for expanding these evaluations include abnormal ECG findings, significant blood pressure changes, arrhythmias detection, changes in heart rate, and evidence of cardiac stress [51].

The respiratory safety panel focuses on pulmonary function tests, including respiratory rate monitoring, lung volume measurements, airway resistance assessment, and gas exchange evaluation. Expansion of these analyses is triggered by respiratory rate abnormalities, changes in lung compliance, signs of bronchoconstriction, hypoxia detection, and respiratory distress symptoms [52].

The neurologic safety panel evaluates neurologic function through motor activity monitoring, sensory function assessment, cognitive performance tests, and behavioral observations. The necessity for comprehensive neurological examinations arises in cases of altered motor function, changes in sensory perception, cognitive impairments, behavioral abnormalities, and seizure activity detection [53].

Integrated safety monitoring involves common triggers for expanded evaluation, such as unexpected adverse reactions, dose-dependent effects, species-specific responses, and clinical relevance indicators [50].

The incorporation of these diverse evaluation methods provides a comprehensive understanding of cellular product biosafety. By combining general toxicity and safety pharmacology analyses, researchers can develop a holistic view of potential risks and benefits associated with cell-based therapies. 

## 3. Assessment of Oncogenicity, Teratogenicity and Tumorigenicity

### 3.1. Oncogenicity

Oncogenicity refers to the malignant transformation of recipient cells under the influence of donor cells [54]. Oncogenicity studies are designed to evaluate the capacity of cellular drug components to induce tumor transformation in recipient cells and tissues [55]. This comprehensive analysis assesses the potential risk of oncological disease development associated with cell therapy. Stem cell–derived extracellular vesicles (EVs) can deliver miRNAs, lncRNAs, mRNAs, and proteins to recipient cells, reprogramming gene expression toward oncogenic pathways, and drug resistance [56]. EVs from adipose-derived MSCs carrying lncRNA NEAT1 enhanced proliferation, migration, and gemcitabine resistance in pancreatic cancer models by modulating the miR-491-5p/Snail/SOCS3 axis, illustrating a direct, transferable oncogenic mechanism [57]. Reciprocal paracrine loops between stem cells and epithelial cells can activate β-catenin and inflammatory prostaglandin pathways, driving epithelial–mesenchymal transition and generating a cancer stem cell–like niche in surrounding tissues [58]. Disease-modified stromal contexts can skew this paracrine output toward pro-tumorigenic signaling (for example, IL-6/JAK–STAT3 activation in epithelial cells), thereby supporting metastatic traits and therapy resistance [59]. Importantly, any cell-based therapeutic product that lacks proper oncogenicity evaluation is considered potentially oncogenic [41]. Despite advancements in vitro testing, laboratory animals remain a crucial component of oncogenicity research. Similar to tumorigenicity studies, immunocompromised animals are the preferred models. However, according to the FDA, the microenvironment significantly influences tumor development. Therefore, it is recommended to administer cells via the intended route of administration for the target cellular product, rather than relying solely on subcutaneous testing. Histological analysis is typically employed to evaluate results [60]. The duration of post-administration observation in experimental animals is critical for assessing long-term consequences and potential tumor development risks in recipients [55]. In vivo research evaluated the biosafety of UC-MSCs for traumatic brain injury (TBI) therapy and found low tumorigenic potential. Tumor formation was assessed by immunohistochemistry for epidermal growth factor receptor (EGFR) variant III, a glioma-associated EGFR mutation, which was not detected in transplanted cells. Proliferation was examined via proliferating cell nuclear antigen (PCNA), showing no significant UC-MSC proliferation over 12 months, although TBI increased proliferation of neighboring neuroblasts [7]. Emerging technologies offer alternative approaches to oncogenicity assessment. Notably, microfluidic chips—or “organs-on-chips” containing patient-derived cells—show promise for in vitro oncogenicity evaluation. This approach could enable tumor formation and metastasis analysis using patient-specific cells without animal testing. However, this technology remains under development and has not yet been widely implemented in preclinical research [60].

### 3.2. Tumorigenicity

Tumorigenicity describes the ability of transplanted cells to form tumors [61]. Literature sources emphasize the importance of conducting tumorigenicity studies for therapeutic agents containing stem cells, low-dose cells with high differentiation potential, and cells exposed to various chemical and physical modifications [62]. It has been hypothesized that the risk of stem cell tumor formation is largely due to the persistence of undifferentiated populations, since injection of undifferentiated embryonic stem cells typically results in teratoma formation. Particular attention should be paid to cells producing mitogenic and immunomodulatory factors [63]. Tumorigenicity evaluation helps determine the ability of transplanted cells to form tumors. This aspect is especially critical when working with stem cells due to their high proliferative activity [60]. At the preclinical stage, various methods are used to determine tumorigenicity, with the most reliable approach involving subcutaneous administration of cellular products to immunocompromised animals, such as athymic nude mice or rats, animals with suppressed immunity receiving drugs like cyclosporine, azathioprine, imuran, and cyclophosphamide, as well as mice with severe combined immunodeficiency (SCID) [64]. The analysis of tumor development dynamics includes monitoring the latency period of tumor growth, evaluating the frequency of tumor occurrence, measuring tumor volume and mass, assessing the presence of invasive growth and metastases, analyzing the proliferative activity of cells, and evaluating their ability to grow uncontrollably [65]. The observation period is subject to variation. However, the majority of researchers have monitored tumor growth in animals for 10–40 weeks, while the FDA recommends monitoring tumor formation in vivo for 4–7 months [64,66]. Preclinical safety studies of human embryonic stem cell–derived retinal pigment epithelial cells (hESC-RPE) for the treatment of age-related macular degeneration showed absence of malignant growth and migratory properties of subcutaneous injection of transplant in immunocompromised NOG mice and subretinal injections into albino rabbit eyes [67].

### 3.3. Teratogenicity

Teratogenicity represents a specific indicator related to the formation of teratomas, which are particularly characteristic of pluripotent stem cells: iPSCs and embryonic stem cells (ESCs). These teratomas are identified through the presence of immature tissue derived from the three primary germ layers: ectoderm, mesoderm, and endoderm. Histopathological analysis serves as the primary method for their detection. The assessment of teratogenicity is predominantly conducted when the target cell product originates from the pluripotent stem cells derivatives [68]. An indirect method for confirming the absence of pluripotent cells in the target product involves evaluating pluripotency markers, including octamer-binding transcription factor 3/4 (OCT3/4), SRY-box transcription factor 2 (SOX2), homeobox protein NANOG (NANOG), Krüppel-like factor 4 (KLF4), tumor rejection antigen 1-60 (TRA-1-60), tumor rejection antigen 1-81 (TRA-1-81), and stage-specific embryonic antigen-4 (SSEA-4) [69,70]. However, the classical in vivo teratoma formation test remains the only definitive method for analyzing the presence of residual pluripotent cells in the cell product. The analysis is typically performed using immunocompromised animals, with various administration routes employed, including subcutaneous, intramuscular, and direct organ injection into tissues such as kidneys, liver, and testes. The evaluation period can range from one week to several months, depending on the specific study parameters [70,71].

In research on Parkinson’s disease cell therapy, striatal transplantation of iPSC-derived dopaminergic neuron progenitor cells (DNPCs) into immunodeficient mice showed no malignant changes on H&E staining and no antigen Kiel 67 (Ki67)-positive proliferation after 52 weeks; subcutaneous teratogenicity testing with DNPCs in collagen likewise yielded no tumors over 26 weeks [39]. A subsequent study employing brain-slice IHC (Ki67, Sox1, Pax6) reported absence of these markers by day 274, while deliberate admixture of undifferentiated iPSCs (1% or 10%) with DNPCs produced teratomas, underscoring the necessity of complete differentiation to mitigate teratogenic risk [72]. Additional work defining teratogenic thresholds found that adding 0.1% embryonic stem cells did not induce teratomas, as opposed to 1% and 10%. Ki67-based proliferation in transplanted cells declined from roughly 25% during the first three months to about 6% by day 266, and histopathology at days 30 and 180 showed no tumors, vascular invasion, or lesions [73]. In amyotrophic lateral sclerosis (ALS) models, transplantation of astrocyte progenitor cells with absent pluripotency markers in vitro (SSEA-4, epithelial cell adhesion molecule (EPCAM), TRA-1-60) showed no tumor formation in brain or spinal cord at 4, 17, and 39 weeks post-transplantation, supporting a favorable biosafety profile [74]. Also, Schwartz S.D. and colleagues demonstrated no evidence of tumorigenicity, teratoma formation, or spread to other body parts at subretinal transplantation of hESC-RPE into immune-deficient mice [75].

In conclusion, thorough assessment of oncogenicity, tumorigenicity, and teratogenicity is essential to ensure the biosafety of cell-based therapies, with particular emphasis on controlling undifferentiated stem cell populations and employing comprehensive preclinical testing.

## 4. Assessment of Immunogenicity in Cell Therapy

Immunogenicity assessment in cell therapy represents the study of immune system reactions that occur in response to cell transplantation. This parameter significantly affects both the biosafety profile and therapeutic efficacy of the cellular product. With stem cell therapy, adverse immune responses range from local inflammation to systemic reactions. Clinical significance includes loss of cell persistence, decreased efficacy, infusion reactions, cytokine-mediated toxicity, rejection of transplant, and the need for immunosuppression; mitigation varies by product and indication [76]. Transplanted cells activate pattern recognition receptors on macrophages, dendritic cells, and NK cells via injury-associated molecular patterns, and glycan patterns, initiating inflammation and shaping adaptive priming [77]. The intensity of HLA mismatch and HLA class II expression have been identified as critical determinants of T-cell-mediated rejection. Research has demonstrated that an escalation in class II mismatch can amplify immune responses, a finding that has been substantiated in the context of homologous tissue implants and that has been extrapolated to the domain of cell transplants [78,79]. The study showed that primary cholangiocyte organoids (PCOs) were tested in co-cultures and humanized mice, revealing that inflammatory cues upregulate HLA-I and HLA-II, which drive allogeneic immune responses mitigated by HLA matching. In vivo, allogeneic PCOs underwent progressive rejection, whereas autologous PCOs caused only low-level infiltration, likely influenced by culture-acquired mutations, cell viability, and matrix factors [80].

### 4.1. Analyzing the Innate Immune Response

The initial step in immunogenicity assessment involves analyzing the innate immune response through precise measurement of cytokine production and granzyme release from NK cells. This evaluation is performed using enzyme-linked immunosorbent assay (ELISA) techniques, which provide quantitative data on the levels of these critical immune mediators. An important aspect of NK cells activity evaluation is the study of degranulation processes through co-culturing NK and target cells. Among the critical indicators for assessment, particular attention is paid to the lysosome-associated membrane protein LAMP-1 (CD107). Thus, it was shown that LAMP 1 can be used as a marker of NK cell degranulation, in particular, suppression of LAMP1 causes inhibition of NK cell cytotoxicity due to the inability to deliver granzyme B to target cells [81]. Additionally, the cytotoxic activity of NK cells against target cells can be quantitatively measured in vitro by assessing lactate dehydrogenase release, as confirmed by recent research [82,83].

The analysis of T-cell activation traditionally relies on the mixed lymphocyte culture reaction (MLR) method. The implementation of MLR requires two components for assessing cell immunogenicity: peripheral blood mononuclear cells (PBMCs) extracted from the recipient’s blood and the tested cells intended for therapeutic use. During this process, the interaction between these components is observed through co-culture system. A key indicator of the cell product’s immunogenicity is the proliferative activity level exhibited by PBMCs when co-cultured with target cells. Higher proliferative activity signifies greater immunogenic potential of the cell product [84]. The immune response can be further evaluated by monitoring the expression of specific cellular markers associated with activated CD4+ and CD8+ T lymphocytes, particularly focusing on CD25 and CD45 markers [85]. In practical applications, T-cell activation assessment often overlaps with the evaluation of the immunomodulatory function of cell therapy. The study showed that a modified MLR assay was applied to evaluate the immunogenicity of iPSC-derived neural progenitor cells (NPCs), comparing autologous and allogeneic responses in parallel. Another study found that NPCs from different sources exhibited immunomodulatory properties, and the origin of the pluripotent cells did not significantly affect the immunogenicity of the NPCs [86].

The complement system plays a critical role in the immune response to cell-based therapies, making complement activation assays essential for monitoring safety and efficacy in cell therapy applications [87]. Anaphylatoxins C3a and C5a are key components in triggering inflammatory responses, modulating immune cell activity [88]. These assays are particularly important for assessing the potential for immune-mediated destruction of therapeutic cells. Complement activation assays are used in several critical areas of cell therapy, including monitoring the immune response to transplanted cells, evaluating the safety profiles of cell products, assessing potential immune-mediated cytotoxicity, investigating cell survival mechanisms, and monitoring therapeutic efficacy [89].

The application of ELISA methodology in immunogenicity analysis is not limited to the assessment of innate immunity. The study showed that, using this method, pluripotent cells exhibited higher expression of the chemokines C-C motif chemokine ligand 5 (CCL5) and C-X-C motif chemokine ligand 12 (CXCL12) than MSCs and NPCs [68]. In addition, it was demonstrated that the secretome of neural crest stem cells was profiled to quantify interferon-gamma (IFN-γ), interleukin-1β (IL-1β), interleukin-2 (IL-2), interleukin-6 (IL-6), interleukin-10 (IL-10), interleukin-12p70 (IL-12p70), and tumor necrosis factor-alpha (TNF-α), enabling assessment of both inflammatory responses and overall immunogenicity [90]. Exposure to inflammatory cytokines significantly impacts immune responses through modulation of HLA expression. The mechanisms through which cytokines influence HLA expression involve several key pathways, including direct activation of HLA gene promoters, induction of signaling cascades, regulation of transcription factors, epigenetic modifications, and alteration of protein synthesis pathways [91]. Numerous major cytokines are involved in HLA regulation. IFN-γ is the primary inducer of HLA class I and II expression [92,93]. IL-6 modulates HLA expression patterns [94]. Type I interferons influence HLA class I expression [95].

Another method for assessing the immune response is Enzyme Linked Immuno-Spot (ELISpot), which is an effective method for measuring the production of cytokines by immune cells at the level of individual cells. The popularity of this analysis has increased dramatically in recent years as researchers try to better understand immune responses in various applications. This assay is particularly relevant for repeated dosing regimens and allogeneic cell products [96].

### 4.2. HLA Typing

Another crucial factor for a complete investigation of cell immunogenicity is the analysis of HLAs expressed on the cell surface. HLA typing represents the process of donor-patient matching based on the HLA system [97]. This process is commonly carried out using a well-established method known as polymerase chain reaction (PCR). The study demonstrated that the qRT-PCR technique was employed to assess HLA expression in MSCs that were under consideration for TBI therapy. The findings revealed the absence of HLA-DRA1, while HLA-DPA1 and HLA-DQA1 were detected. Notably, the cycle threshold values for MSCs were approximately 32 cycles, whereas for PBMCs, they were around 24 cycles. The authors interpreted this discrepancy as an indication of immunological tolerance [7]. Meanwhile, the active development of technologies makes it possible to select patients using various types of sequencing [98]. Another option for HLA typing is the lymphocytotoxicity test, which operates on the principle of forming an antigen–antibody complex during the interaction between serum containing antibodies to HLAs and the patient’s lymphocytes [98]. Donor-specific anti-HLA antibody monitoring is another method of compatibility assessment focusing on immunoglobulins produced by the recipient’s immune system against donor HLAs [99]. These antibodies play a pivotal role in transplant rejection and determining compatibility. The key objectives of monitoring include transplant rejection risk assessment, evaluation of compatibility between donor and recipient, surveillance of immune response post-transplantation, and detection of sensitization to donor antigens. The clinical significance of anti-HLA antibodies is substantial, impacting transplant outcomes in various ways. They increase the risk of acute rejection, reduce graft survival rates, necessitate adjustments in immunosuppressive therapy, and may lead to humoral rejection [100].

Autologous cells offer a distinct advantage by minimizing the risk of immunological rejection. However, this approach presents several challenges. The process involves isolating cells from the patient’s own biological material, followed by an extended period of in vitro cell cultivation. These procedures are characterized by high labor intensity, significant time requirements, and considerable financial costs. At the same time, allogeneic cells make it possible to form a bank for cell therapy, which, in turn, allows scaling up the production process and minimizing the time spent on manufacturing the cell product. Moreover, the use of allogeneic cells enables the organization of a reliable biosafety verification system for the target product. However, these types of cells can cause an excessive immune response from the recipient’s body [39,101,102].

It is worth emphasizing that different types of cells have different immunogenic properties. Thus, MSCs have low immunogenicity, since they lack HLA class II, which suppresses CD4+ T cells and provides immunomodulatory properties, which is of great importance for allogeneic transplantation [103]. Other types, such as HSCs, neural stem cells, and others, have more pronounced immunogenicity and are able to induce a significant immune response after transplantation [104]. Currently, methods are being developed to overcome these limitations using differentiation into the desired type of autologous iPSCs, but there remains the possibility of an immune response due to reprogramming errors [105]. On the other hand, approaches to creating hypoimmunogenic or "universal" stem cells by knocking out MHC genes using CRISPR and other modifications, such as those that target HLA-DRA1, are actively being introduced. These cells often employ specific strategies to avoid NK cells recognition, primarily through HLA-E overexpression, which binds to inhibitory receptors on NK cells, preventing their activation, and HLA-G overexpression, known for its immunosuppressive properties, binding to multiple inhibitory receptors on immune cells [106].

In conclusion, immunogenicity is a critical determinant of cell therapy biosafety and efficacy, necessitating comprehensive assessment of innate and adaptive immune responses to inform the development of both allogeneic and autologous products.

## 5. Cell Distribution Assessment

Biodistribution refers to the characterization of how administered therapeutic cells migrate, localize, persist, and accumulate within different tissues and organs within the recipient over time [107]. Following the local administration of the transplant, the cell may migrate to other tissues and organs. Such migration throughout the body increases the risk of adverse reactions in the patient [108]. The study of cell distribution within the body is of paramount importance in the context of cell therapy, as it directly influences the therapeutic efficacy, biosafety, and predictability of outcomes [109]. Tissue localization is essential for determining whether therapeutic cells reach their intended targets and localize appropriately. Assessment of persistence and clearance provides insight into how long cells remain in the body and the mechanisms of their elimination, informing both therapeutic duration and potential long-term risks [110]. Analyzing time-dependent distribution, or changes in cell localization over time, offers a dynamic perspective on cellular behavior and therapeutic efficacy, supporting the development of safer and more effective interventions [111].

### 5.1. Cell Labeling and Visualizing

Cell labeling is one of the main methods applicable for monitoring the migration of target cells during preclinical research, with the selection of labels being determined by the research objectives and visualization techniques utilized. Based on one classification system, two primary types of labels are recognized: direct and indirect [112]. Direct labels encompass various nanoparticles and chemical reagents. These types of markers are particularly effective when conducting biodistribution studies over short-term periods (several days), as prolonged use may result in label loss due to cell proliferation processes. Indirect labels involve genetically introduced reporter genes [113]. Their advantage lies in their prolonged retention within cells, enabling extended observation periods during research. For effective distribution analysis, labels must meet several criteria: complete cellular compatibility (non-toxicity), safe elimination from the body post-cell death, sufficient stability for experimental duration, consistent retention within the cell for the required experimental period [114,115]. The method of visualizing labeled cells is also very important. Some of the most promising tools are positron emission tomography (PET) and single-photon emission computed tomography (SPECT). To obtain results with PET and SPECT, cells must be pre-labeled with radioisotopes [115]. Another method of visualizing cell biodistribution is magnetic resonance imaging (MRI). To work with this technology, target cells must be labeled with paramagnetic iron oxide particles or other similar markers. MRI allows for precise assessment of cell distribution in soft tissues at high resolution, with the generation of a three-dimensional image [115,116]. However, it is important to note that besides the aforementioned methods, there are alternative approaches for evaluating cell biodistribution.

### 5.2. PCR and Histology Methods for Cell Detection

The PCR method can be employed, in which the organs of the animal are analyzed separately at a certain time interval following cell injection. Currently, this method stands out as one of the most widely used and straightforward techniques in preclinical research implementation. PCR allows sensitive detection of transplanted cells by amplifying donor-specific genetic markers, enabling identification even when cell numbers are very low. This method provides quantitative information on cell distribution across tissues, offering high specificity and reproducibility compared to histological techniques [117]. Immunohistochemistry represents another commonly employed method, where tissue and organ sections from animals previously injected with target cells undergo antibody labeling against specific markers characteristic of the investigated cells [115]. In preclinical research, in situ hybridization technology is also used to assess cell distribution. This is a method employed to detect and localize specific nucleic acid sequences in tissue sections [118].

An example is a study on the biodistribution of oligodendrocyte progenitor cells (OPCs) after direct injection into the traumatized area of the spinal cord. The biodistribution of transplanted cells was systematically evaluated at intervals up to nine months post-transplantation in a rat model of cervical spinal cord injury. Histology, in situ hybridization, and PCR of blood, cerebrospinal fluid, and tissue samples were employed to trace the transplanted cells, while extensive histopathological examination of brain and spinal cord sections was performed to assess potential migration. These methods revealed no dissemination to peripheral organs, with cell migration largely confined to the cervical region and limited rostral or caudal extension [119]. In addition, another study demonstrated the distribution of a novel ALS therapy based on astrocyte progenitor cells (APCs) following intrathecal injection into the cerebrospinal fluid of immunocompromised NSG mice. Cell localization within the central nervous system was assessed over 39 weeks using in situ hybridization targeting the Alu Y sequence, revealing widespread but non-uniform distribution, with higher concentrations along the meninges. Extra-central nervous system biodistribution was evaluated by real-time PCR in multiple organs, including spleen, liver, kidneys, heart, lungs, bone marrow, and reproductive tissues, with no significant human DNA detected, confirming the absence of peripheral migration [74]. Moreover, the distribution of DNPCs in a Parkinson’s disease model was evaluated by injecting them into the striatum of immunodeficient rats and analyzing tissues at 26-, and 39 weeks post-injection. PCR-based assessment revealed no migration beyond the cerebral hemispheres [120]. The main approaches to assessing the biosafety of cell products are presented in Figure 1.

In summary, the study of biodistribution is essential for ensuring the biosafety, efficacy, and predictability of cell-based therapies. A broad spectrum of labeling and visualization techniques, ranging from molecular approaches to advanced imaging modalities, provides researchers with the tools necessary to track cellular fate and optimize therapeutic strategies.

## 6. Quality of Cellular Products

The quality of biomedical cellular products is a critical aspect that determines their safety, efficacy, and consistency. Comprehensive quality assessment is required at all stages of development and production to ensure successful clinical application. The quality control system includes the evaluation of both production processes and the consistency of product characteristics [121]. The main components of quality assessment start with identity and authenticity, which involves morphological characteristics such as cell shape, size, and structure, as well as specific marker expression, including surface markers. Additionally, the gene expression profile and protein expression are analyzed to ensure the authenticity of the cell product. Senescent markers are also assessed to verify the consistency of the cell line [122].

Purity is paramount in the evaluation process. This includes sterility testing to confirm the absence of microbial contamination, pyrogenicity assessment to ensure the lack of fever-causing agents, and mycoplasma control to verify the absence of contamination [123].

Biological activity is another crucial aspect, encompassing cell viability evaluation, proliferative capacity, functional activity, and differentiation potential. Safety evaluation focuses on genetic stability through karyotype analysis, chromosomal abnormality detection, genetic instability assessment, and malignancy potential evaluation [124].

Testing methods employed in quality control include laboratory analysis through flow cytometry, DNA fingerprinting, STR profiling, cytogenetic analysis, and molecular biology techniques [125,126].

The required time to produce cellular products is a critical factor, particularly in therapies for rapidly progressing diseases. Cellular products, especially CAR-T or CAR-NK, demand significant time for collection, engineering, and quality control. Minimizing production time is essential for patient safety, as extended waiting periods can result in accelerated disease progression.

To balance product quality and availability within a clinically relevant timeframe, several factors must be considered. The use of cryopreservation or banking methods for temporary storage of cellular products can reduce certain production stages [127]. Implementing standardized protocols minimizes time spent at each stage, including the use of validated methodologies and automated control systems. The use of robotic systems, which reduce the risk of culture contamination by eliminating human contact, maintains optimal conditions for cell growth, such as temperature, humidity, carbon dioxide, and oxygen concentrations. This approach lowers the risk of errors and accelerates processes [128].

The study found that human epidermal stem cells (EpiSCs) remained stable from passage 1 (P1) to P8, as determined by the evaluation of specific markers, telomerase activity, and cellular senescence. However, over this period, cell senescence increased while telomerase activity decreased [129]. In another preclinical study, the quality of the cell product was assessed using the expression of specific BMSCs markers (CD105, CD90, and CD73), the ability to adipogenic and osteogenic differentiation, karyotyping and the absence of contamination of bacteria, fungi, mycoplasma, hepatitis virus, and endotoxins [130].

In summary, proper quality control of cellular products ensures their safety, efficacy, and consistency, which is critical for successful clinical application and positive patient outcomes. Continuous monitoring and adherence to strict quality standards are essential for the development of reliable and effective biomedical cell products. Criteria used to assess the safety of cellular products are summarized in Table 1.

## 7. Regulatory Requirements for Biosafety Assessment of Cell Therapy

The regulation of cell therapy biosafety represents a critical component of global biomedical oversight, ensuring that these innovative treatments, which harness living cells to repair, replace, or regenerate damaged tissues, are both efficacious and safe for patients. Regulatory agencies such as FDA, EMA, and Japan’s Pharmaceuticals and Medical Devices Agency (PMDA) have developed frameworks grounded in shared scientific principles while incorporating nuanced differences tailored to their respective legal and cultural contexts [131]. At the heart of these frameworks is a risk-based approach, which calibrates the stringency of requirements to the inherent complexities of cell therapies, including their autologous or allogeneic origins, degrees of manipulation, and potential for long-term effects such as tumorigenicity or immunogenicity. This approach facilitates adaptive regulation, acknowledging the heterogeneity of cell products and the need for tailored data on quality, biosafety, and efficacy [132].

Core principles common to the FDA, EMA, and PMDA emphasize robust quality and manufacturing controls (Chemistry, Manufacturing, and Controls or CMC) to verify the identity, purity, potency, and consistency of cell therapies from raw materials to final product. This includes stringent measures to mitigate contamination risks, such as microbial and viral agents. Non-clinical studies are mandated to establish proof-of-concept, safe dosing, administration routes, and potential toxicities. Clinical trials must prioritize patient protection, generating reliable data on biosafety and efficacy through ethical designs. Long-term follow-up is required to monitor delayed adverse events, given the persistence of cells in the body. Finally, traceability systems ensure bidirectional tracking from donor to recipient, supporting pharmacovigilance. Biosafety assessment methods for cell therapies span in vitro, in vivo, and clinical domains, forming a layered evaluation to identify and mitigate risks comprehensively [133]. In vitro methods are foundational, particularly within CMC and non-clinical phases, and include assays for sterility to detect bacterial and fungal contaminants, mycoplasma testing via culture or PCR-based techniques, and human pathogen screening using PCR for viruses such as human immunodeficiency virus, human T-lymphotropic virus, hepatitis B and C, cytomegalovirus, Epstein–Barr virus, and parvovirus B19. Adventitious virus testing employs cell line co-cultures (e.g., human diploid, monkey kidney, or product-specific lines) to reveal unexpected contaminants, supplemented by high-throughput sequencing (NGS) for broad detection and transmission electron microscopy for virus particle visualization. Retroviral and species-specific virus testing address risks from non-human feeders or animal-derived materials, while residual vector quantification ensures minimal carryover from genetic modifications [134,135,136]. Whole genome sequencing detects mutations, off-target edits, or integrations in genome-edited or continuous cell lines, and cytogenetic testing (e.g., G-banding) confirms karyotype normality in expanded primary cells [134,137].

Clinical methods shift the focus to human subjects, emphasizing biosafety and tolerability in early-phase (first-in-human) trials through intensive monitoring of adverse events, staggered enrollment, and predefined stopping rules. Dose-finding studies determine safe and effective ranges based on non-clinical data. Pharmacodynamics assessments measure in vivo responses, such as gene expression or immune activation, to confirm mechanisms [138]. Long-term follow-up, often extending to 15 years for gene therapies, detects delayed risks like tumorigenicity or infections while verifying efficacy durability. Holistic risk assessments encompass the entire procedure, including surgical administration, pre-treatments (e.g., lymphodepletion), and concomitant medications [139].

Post-marketing surveillance includes monitoring of side effects, long-term biosafety assessment and analysis of efficacy in real-life conditions. International regulators allow accelerated registration of products intended for the treatment of severe and life-threatening conditions, while establishing exemptions from registration for cellular products of personalized therapy, hematopoietic stem cells and blood products [140].

While the FDA, EMA, and PMDA converge on these principles and methods, differences arise in regulatory formality, approval pathways, and emphases. The FDA’s guidance expedited programs (e.g., Fast Track, Breakthrough Therapy, Regenerative Medicine Advanced Therapy) enable rolling reviews and accelerated approvals based on surrogates, with post-approval confirmatory trials. In contrast, the EMA’s guidelines are rooted in binding EU regulations (e.g., Regulation (EU) No 536/2014), providing a more prescriptive framework. The EMA strongly promotes the 3Rs in non-clinical studies, encouraging alternatives to animal testing, and allows accelerated assessments but prioritizes confirmatory trials for marketing authorization, emphasizing deviations from conventional testing must be justified. The PMDA’s administrative notices serve as reference materials without legal status, yet they uniquely integrate a conditional and time-limited approval pathway for regenerative products, addressing heterogeneity and medical needs through mandatory post-marketing efficacy plans that detail sample sizes, controls, and data collection for full approval—a more structured post-market efficacy requirement than the FDA’s or EMA’s biosafety-focused surveillance. Additionally, the PMDA provides detailed guidance on autologous product challenges, such as donor variability and limited sampling, necessitating agency consultations, while the FDA exempts autologous therapies from certain allogeneic donor rules, and the EMA focuses on comprehensive component characterization [134,135,136].

## 8. Current Limitations and Future Direction

Despite substantial progress in enhancing the biological biosafety of cell therapies, a number of critical limitations persist. The inherent biological variability of cell products poses significant challenges for quality control procedures and subsequent risk assessment. For instance, the challenge of comprehensively characterizing cell cultures and the significant heterogeneity that characterizes them persist as major obstacles [141]. Furthermore, challenges arise in the standardization of storage protocols, such as cryopreservation and the manufacturing of living cell-based products. This encompasses culture media and excipients present in the finished product [142]. The existence of regional discrepancies in the domains of pre-release testing, efficacy testing, and post-marketing surveillance represents a substantial impediment to the execution of multicenter clinical trials and the subsequent commercialization of products. There is an urgent need to harmonize standards for genomic stability assessment, immunogenicity testing, and long-term follow-up protocols [140]. Moreover, a paucity of extensive long-term biosafety data exists for a considerable number of new cell therapies. While post-marketing registries and FDA’s Adverse Event Reporting System (FAERS) analyses offer valuable insights, the true incidence of late events, such as secondary malignancies or chronic immune dysregulation, will only become apparent with long-term follow-up. The establishment of robust international biosurveillance networks, characterized by standardized data collection and reporting, is imperative to overcome this limitation and ensure long-term patient biosafety [143]. Furthermore, the majority of extant studies have focused on therapeutic efficacy, while only a few have comprehensively examined biosafety.

Risk-based approaches to drug development are becoming increasingly important as regulators have come to understand that traditional paradigms may be inadequate for complex biological products. These approaches emphasize the importance of understanding the risks specific to a particular product and implementing appropriate strategies to minimize them throughout the product life cycle [144]. The development and implementation of accelerated approval and marketing authorization processes is needed to facilitate accelerated access to promising cell-based therapies while maintaining biosafety standards [145]. These regulatory mechanisms would facilitate earlier market entry based on surrogate endpoints while requiring comprehensive post-marketing studies to confirm biosafety and efficacy. Additionally, collaborative efforts among federal agencies in multiple nations are imperative to establish consistent biosafety assessment protocols.

Further development of technology for standardization of cell-based drug production is needed. One of the central approaches involves the establishment of robust and reproducible manufacturing protocols that minimize batch-to-batch variability and ensure consistent product quality. This includes the refinement of upstream processes such as cell sourcing, expansion, and differentiation, as well as downstream steps such as purification, formulation, and cryopreservation [146]. Standardization requires the integration of advanced bioreactor systems capable of providing tightly regulated culture conditions, including oxygen levels, nutrient supply, and shear stress, to maintain cell viability and functionality [147]. In addition, improvements in technologies for comprehensive biosafety assessment are required, including new in vitro models based on organoids, organs on a chip, and microfluidic systems. These models provide physiologically relevant microenvironments that mimic human tissue architecture, cellular interactions, and dynamic signaling pathways more accurately than traditional two-dimensional cultures [148]. The use of modern omics technologies can also contribute to more effective assessment of both efficacy and biosafety by enabling comprehensive, systems-level analyses of therapeutic cells and their interactions with host tissues. Genomics and epigenomics provide insights into genetic stability, mutations, and chromosomal rearrangements that may arise during cell expansion or gene editing [149]. Transcriptomics and proteomics allow the characterization of gene expression patterns, signaling pathways, and protein networks associated with therapeutic activity or unintended differentiation [150]. Metabolomics and lipidomics, in turn, reveal shifts in cellular metabolism that may influence cell functionality, persistence, or immunogenicity. Integration of multi-omics data through advanced bioinformatics approaches facilitates the identification of predictive biomarkers of efficacy and biosafety, as well as the development of standardized assays for routine quality control [151].

Using optogenetic and chemogenetic tools to control cell behavior presents a promising avenue for advancing biosafety studies. These innovative approaches offer precise methods to monitor and regulate cellular activity, which is crucial for ensuring the safety and efficacy of biological products [152]. In biosafety research, optogenetic tools provide real-time control over cellular processes, enabling researchers to monitor cell distribution and migration patterns, assess potential off-target effects, study cell-to-cell interactions, and evaluate proliferation rates under controlled conditions. Chemogenetic tools, with their sustained and reversible effects, allow for long-term observation of cellular behavior, controlled activation of specific pathways, assessment of long-term safety profiles, and study of chronic effects on host tissues. Tracking the distribution and activity of cells using genetic barcodes and liquid biopsy methods provide valuable insights into cell behavior, migration patterns, and potential risks associated with cell-based therapies [153]. Genetic barcoding technology involves unique molecular tags inserted into cells, allowing researchers to track individual cell lineages, monitor cell proliferation rates, identify off-target migration, assess long-term cell survival, and detect potential tumorigenic transformation. The key advantages of genetic barcoding include high specificity and sensitivity, the ability to track multiple cell populations simultaneously, long-term monitoring capabilities, and non-invasive detection methods [154].

Liquid biopsy represents a minimally invasive approach to monitor cell distribution and activity through analysis of circulating DNA, detection of cell-free RNA, and identification of extracellular vesicles [155]. The benefits of liquid biopsy in biosafety research encompass real-time monitoring of cell behavior, early detection of adverse events, quantitative assessment of cell presence, and minimal patient discomfort [156].

It is essential to enhance the mechanisms of control over already transplanted cells, including the implementation of suicide switches and encapsulation techniques. Suicide switches and biosafety mechanisms represent a promising approach to enhance the controllability of therapeutic cells [157]. Examples of such technology include lenalidomide-induced switches and CRISPR-based suicide systems. The implementation of these methodologies within clinical practice could contribute to enhancing the biosafety profile of cell therapy [158]. Biocontainment strategies involving cellular microencapsulation present a novel approach to the management of biosafety risks. These technologies facilitate the physical isolation of therapeutic cells while preserving their biological functionality, thereby potentially mitigating immunogenicity and migration risks. The development of contemporary biomaterials designed for cell encapsulation has the potential to enhance the transport of nutrients while preserving immune isolation [159]. The implementation of engineered targeting mechanisms ensures the precise delivery of therapeutic cells to target tissues, thereby minimizing adverse effects. Advanced engineering approaches encompass synthetic receptor systems and programmable cellular circuits that exhibit responsiveness to particular environmental stimuli [160,161].

The prospect of future success in the domain of cell therapy biosafety is contingent upon the sustained collaboration among researchers, manufacturers, regulators, and clinicians, aimed at the development and implementation of comprehensive biosafety strategies. The integration of emerging technologies with established biosafety principles has the potential to transform cell therapy from an experimental treatment to a mainstream therapeutic modality with well-characterized and manageable biosafety profiles. As the field continues to mature, it is imperative to emphasize proactive biosafety design and continuous improvement to maintain public trust and ensure the sustainable development of this transformative therapeutic approach (Figure 2).

## 9. Conclusions

Regenerative medicine offers promising opportunities for the treatment of previously untreatable diseases, yet the biosafety of cellular therapies remains a central challenge. Advances in assessment methods, including immunological, molecular, and genetic approaches, have significantly improved the ability to evaluate risks and ensure product biosafety. Ongoing refinement of these strategies, alongside strict regulatory oversight, will be crucial for the successful and safe integration of cellular therapies into clinical practice.

## Figures and Tables

**Figure 1 cells-14-01660-f001:**
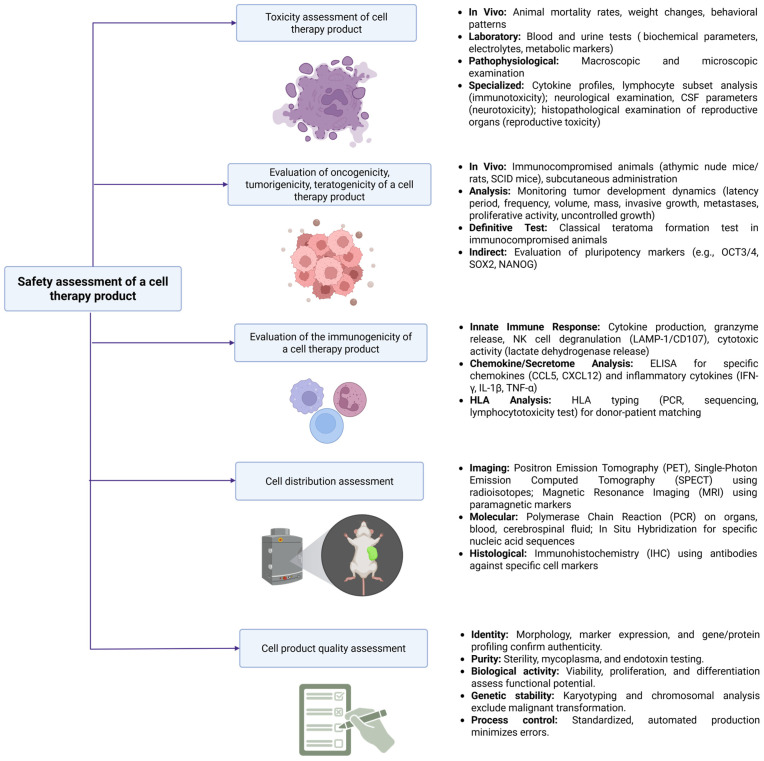
The main approaches to the biosafety assessment of cell products and the corresponding methods.

**Figure 2 cells-14-01660-f002:**
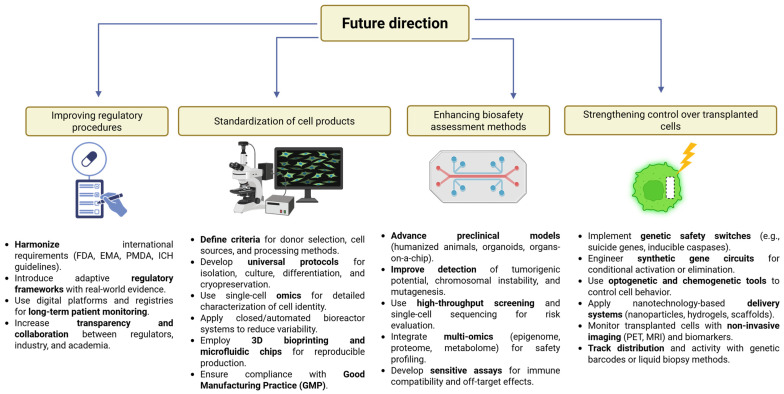
Potential directions for improving the quality of cellular product biosafety assessment.

**Table 1 cells-14-01660-t001:** Examples of biosafety studies of cell products.

Cell Type	Disease	Method of Cell Administration	Toxicity	Tumor Formation	Cell Distribution	Other	Reference
DNPCs (IPSCs)	PD	Brain’s striatum of NOG mice (2 × 10^5^ cells per mouse); subcutaneous in NOG mice (6 × 10^5^ cells); stereotactic injection of 4 × 10^5^ cells in nude rats	Animal survival rate, general condition, behavior, blood and urine tests; pathohistological analysis (52 weeks);	Histology: markers of DNPCs (FOXA2 and TUJ1), markers of iPSCs (OCT3/4, LIN28, and TRA-2-49), markers of early neuronal progenitor cells (SOX1, PAX6); proliferation marker (Ki67), H&E staining;	Histology: H&E, IHC (Ku80);	Proliferative activity: histology (Ki67); cell survival: histology	Doi D. et al., 2020 [39]
DNPCs (IPSC)	PD	Brain’s striatum of the immunodeficient mice (1 × 10^5^ cells)	Survival, animal weight, blood analysis, histopathology;	Histology: pluripotency marker (Oct4), proliferation marker (Ki67), H&E staining;	PCR;	Proliferative activity: histology (Ki67, Sox1, and Pax6); Survival of transplanted cells: histology (hCAM);	Jeon J. et al., 2025 [72]
APCs	ALS	Intrathecal injection of cells into the cerebrospinal fluid of transgenic hSOD1 mice (3 × 10^6^ cells per mice)	Survival, animal weight, blood analysis, histopathology;	Histopathology, histology: pluripotency markers (SSEA-4, EPCAM, and Tra-1-60);	In situ hybridization (Alu Y sequence), PCR;	Proliferative activity: histology (Ki67); Survival of transplanted cells: histology (Stem121, Stem123);	Izrael M. et al., 2018 [74]
DNPCs (ESC)	PD	Intrastriatal injection of cells into immunodeficient mice (4 × 10^5^ cells)	Survival, animal weight, blood analysis, histopathology;	Histology: markers of DNPCs (FOXA2, LMX1A, and PITX3), pluripotency marker POU5F1; H&E staining + histopathology (groups: DNPCs+ 0.1% ESC, DNPCs+ 1% ESC, DNPCs+ 10% ESC, 100% ESC)	PCR;	Proliferative activity: histology (Ki-67) Cell survival: histology (STEM121)	Piao J. et al., 2021 [73]
DNPCs (ESC)	PD	Intrastriatal injection of cells into immunodeficient mice (7 × 10^5^ cells)	Survival, animal weight, blood analysis, body temperature, appetite, behavioral tests (Irwin test), histopathology, organ weights	Histopathology	PCR;	Proliferative activity: histology (Ki67); Survival of transplanted cells: histology (hNCAM)	Kirkeby A. et al., 2023 [120]
UC-MSCs	TBI	In situ (5 × 10^4^ cells) or into the tail vein (5 × 10^5^ cells)	-	Pathohistology: mutation variant of epidermal growth factor receptor type III—EGFRvIII	Fluorescent visualization (UC-MSCs labeled with CFSE);	Proliferative activity: PCNA; Immunomodulatory	Wang G. et al., 2022 [7]
OPCs	Cervical spinal cord injury	Injection into spinal cord parenchyma of nude immunodeficient rats (2.4 × 10^5^ or 2.4 × 10^6^ cells per rat)	Survival, animal weight, blood analysis, grooming; general state, activity, allodynia assessment, histopathology.	Histopathology	In situ hybridization, PCR, histology	Proliferative activity: histology (Ki67);	Manley NC. et al., 2017 [119]
hESC-RPE	Stargardt’s macular distrophy	Subretinally in immune-deficient mice	-	Immunohistochemistry	Fluorescent visualization	Immunophenotyping and karyotyping of hESC-RPE, cell survival: histology	Schwartz et al., 2016 [75]
hESC-RPE	Age-related macular degeneration	Subcutaneous injection in immunocompromised NOG mice (1 × 10^7^ per mice) or subretinal injections into albino rabbit eyes (5 × 10^4^ cells)	-	Histology: H&E staining PCR: expression RPE-specific markers	Multicolor-confocal scanning laser ophthalmoscopy and immunohistochemistry: markers specific for RPE (NuMA and BEST-1)	Immunophenotyping, genotyping, whole-genome sequencing analysis, single-cell RNA sequencing, and karyotyping of hESC-RPE, cell survival: histology	Petrus-Reurer et al., 2020 [67]
EpiSCs	wound repair	Subcutaneously injection in female athymic nude mice (1 × 10^7^ per mouse EpiSCs was injected)	-	Histology: H&E staining	-	Senescence, telomerase activity assay, and transcriptome analysis for assessment of quality of EpiSCs	Zhao et al., 2023 [129]
hBMSCs	-	Intravenous injection into BALB/c mice low (1.0 × 10^6^/kg), medium (1.0 × 10^7^/kg), and high (1.0 × 10^8^/kg) concentrations of hBMMSCs	Survival, animal weight, general state, blood analysis, spontaneous behavior, histopathology	-	-	Tests for bacteria, fungi, mycoplasma, hepatitis virus, and endotoxin; immunophenotyping and karyotyping of MSCs	Liang et al., 2023 [130]

## Data Availability

No new data were created or analyzed in this study.

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
