# Peer review of "Safety Assessment of Stem Cell-Based Therapies: Current Standards and Advancing Frameworks"

_cells, 2025, doi:10.3390/cells14211660_

Round 1

Reviewer 1 Report

Comments and Suggestions for Authors

Leonov and colleagues provide a review article entitled “Safety assessment of stem cell-based therapies: current standards and advancing frameworks.” Their articulated objective is to “provide a comprehensive analysis of critical biosafety issues, identify prevailing challenges in the field, and evaluate the methodologies currently used to assess them.” They cover key biosafety topics including cellular product toxicity (physiologic, immune-related, and reproductive), oncogenicity, teratogenicity, tumorigenicity, immunogenicity, and anatomic distribution. They conclude with a survey of regulatory requirements, barriers to progress, and future directions. This review provides a uniquely comprehensive overview of key biosafety considerations for a broad range of cellular therapies. To this reviewer’s knowledge, no other recent reviews encompass this scope of cell therapy products and biosafety considerations. Combined with the increasing clinical prominence of cell therapies, this review’s breadth of biosafety content distinguishes it as relevant and meaningful. With specific revisions as suggested below, it is likely to be of interest to readers of this journal and has potential to garner considerable citations in the future.

This review functions well as a broad survey of biosafety considerations in the development of cell therapies. The authors have done well to coalesce disparate topics in cell therapy biosafety into a reasonably clear and cohesive whole. There are relatively few objective gaps in relative content. The authors cite relevant and recent original research reports. By my estimation greater than 60% of their citations are from 2020 or later. As such, this review provides unique value as a comprehensive review of recent data in cell therapy biosafety and is likely to be of interest to the readership of Cells.

I do have several constructive suggestions to maximize the use of this review by investigators in the field of cell therapies.

  1. In terms of content, the authors would do well include a reference to allogeneic hematopoietic stem cell transplantation as an archetype of cell therapy that has helped illuminate many of the biosafety issues addressed in this review and through the Center for International Blood and Marrow Transplant Research (CIBMTR) has demonstrated the potential benefits of industry-wide safety and quality standards, reporting, and surveillance. While this review is focused on emerging cell therapies, it would still be appropriate to acknowledge the historical example provided by this more established treatment modality, which highlights the tremendous strides in safety that can result from careful retrospective review, thoughtful harmonization, and adoption of evidence-based standards.
  2. Another biosafety topic that warrants consideration given it clinical relevance and the broad scope of this review is turnaround or manufacturing time. Many engineered cell therapies such as chimeric antigen receptor T-cell or NK-cell therapies used for malignant indications are limited by the time required for collection, engineering, and quality assurance of the cell product (‘vein-to-vein’ time), which in some cases competes with the kinetics of patients’ disease progression. Thus, considerable effort has been dedicated to minimizing ‘vein-to-vein’ time to maximize patient safety by reducing the risk that individuals will experience accelerating disease progressing during the time between apheresis and pre-cell infusion lymphodepleting chemotherapy. While this is admittedly tangential to biosafety per se, it could be useful to mention this in the section on cellular product quality in terms of balancing the imperatives of ensuring product quality and availability within a clinically meaningful time interval.
  3. In several sections of this review, references are made to standardized criteria or scoring systems, which would be helpful to identify. For example, in section 2, on page 4 line 144, the authors state that “Standardized toxicity scoring systems ensure that adverse events are assessed consistently across studies.” A citation is given for another review by Ai and colleagues in the Journal of Hematology & Oncology, but this review focused primarily on engineered immune cell engineering and not on toxicity scoring. For a review like this, it would be most helpful to name or even describe the most commonly used toxicity scoring systems. This could be a good opportunity to package information concisely in a Table. 
  4. Figure 2 is a schematic of potential future directions for optimizing cellular product biosafety. Typically, such schematics represent a concise visual summary of the material discussed in the text. However, there are certain elements in the schematic that are not addressed in the text. For example, under “Strengthening control over transplanted cells,” the following appear only in this schematic and not in the text:
    • Use optogenetic and chemogenetic tools to control cell behavior
    • Track distribution and activity with genetic barcodes or liquid biopsy methods

Additional constructive suggestions regarding figures, tables, and overall organization are provided below:

  1. The review is information-dense and text-heavy. It would improve readability immensely to have more sub-sections with their own headings and tables that summarize the content within each section. For example, section 2 identified several key mechanisms of cell therapy toxicity – immunological responses, tumorigenesis, cellular senescence, and administration-related complications. These could be organizational sub-headings within this section. Section 2 also has a paragraph beginning at line 107 that identifies the goals of toxicity studies and key analytical methods. This content would be much more likely to be used by readers and cited if some of it was packaged into a table with key goals, methods, and examples. Similar suggestions could be made in other sections to help this review achieve greater clarity and impact.
  2. It is helpful to have summaries of the review's content as the authors provide in Figures 1 and 2. However, the visuals add very little value, in some instances seem irrelevant (e.g., the overlaid histograms, the 96-well plate), and the boxes, arrows, and images occupy space that would be more valuable if it allowed the text to be larger. Given the preponderance of text in Figures 1 and 2, this information would be clearer and easier to navigate if it was displayed in the form of tables with major sub-sections.
  3. The table (I cannot tell the page number since those are missing) lacks a legend and is disproportionately focused compared to the breadth of scope in the rest of the review. The sentence referring to this Table in the text says, “The following biosafety criteria are of critical importance in cell-based research and therapy: oncogenicity, teratogenicity, and tumorigenicity (Table 1).” Yet, the Table conveys information related to the following categories: Toxicity, Tumor Formation, Cell Distribution, and Other. Additionally, unlike the rest of the review and even this section, it is focused exclusively on cell therapies for neurologic diseases. This Table would be much more effective if it aligned more clearly with what is stated in the text, if it had a legend to explain it, and if the rationale for the specific focus on neurological disorders was provided.
  4. The titles for sections 2 and 3 are identical. I suspect one of them is mistaken. This is another opportunity for clearer organizational separation of the dense content of this review.
  5. There are examples of cell products being named but not described. For example, on page 2, line 58, there is a reference to rexlemestrocel-L within this sentence, “Another one cell-based drug approved by the FDA is Rexlemestrocel-L which is aimed at therapy of several congenital heart defects.” Given the broad scope of this review, it would be advisable to be specific as to the nature of a product the first time it is mentioned. The sentence given above could be, “Another cell-based drug approved by the FDA is the allogeneic bone marrow-derived mesenchymal precursor cell product Rexlemestrocel-L, which is labeled for the treatment of congenital heart defects.” 

Overall, this is a relevant, timely, and information-dense review, but it could be much more readable and impactful with improved organization of the text and optimization of the Figures and Table.

Author Response

We are grateful for your time and effort spent reviewing our manuscript. We highly appreciate

your constructive feedback and thoughtful comments. We revised the manuscript according to your comments. All changes were highlighted in yellow color.

Comment 1. In terms of content, the authors would do well include a reference to allogeneic hematopoietic stem cell transplantation as an archetype of cell therapy that has helped illuminate many of the biosafety issues addressed in this review and through the Center for International Blood and Marrow Transplant Research (CIBMTR) has demonstrated the potential benefits of industry-wide safety and quality standards, reporting, and surveillance. While this review is focused on emerging cell therapies, it would still be appropriate to acknowledge the historical example provided by this more established treatment modality, which highlights the tremendous strides in safety that can result from careful retrospective review, thoughtful harmonization, and adoption of evidence-based standards.

Response 1. Thanks for your comment. We have added information about the problems encountered during the first hematopoietic stem cell transplantation and how these problems were circumvented. The information has been added to the introduction section. 

Comment 2. Another biosafety topic that warrants consideration given it clinical relevance and the broad scope of this review is turnaround or manufacturing time. Many engineered cell therapies such as chimeric antigen receptor T-cell or NK-cell therapies used for malignant indications are limited by the time required for collection, engineering, and quality assurance of the cell product (‘vein-to-vein’ time), which in some cases competes with the kinetics of patients’ disease progression. Thus, considerable effort has been dedicated to minimizing ‘vein-to-vein’ time to maximize patient safety by reducing the risk that individuals will experience accelerating disease progressing during the time between apheresis and pre-cell infusion lymphodepleting chemotherapy. While this is admittedly tangential to biosafety per se, it could be useful to mention this in the section on cellular product quality in terms of balancing the imperatives of ensuring product quality and availability within a clinically meaningful time interval.

Response 2. Thanks for the comment. Information regarding required time for manufacturing cellular products to reduce patient waiting time to receive treatment was added in new section “Quality of cellular products”. 

Comment 3. In several sections of this review, references are made to standardized criteria or scoring systems, which would be helpful to identify. For example, in section 2, on page 4 line 144, the authors state that “Standardized toxicity scoring systems ensure that adverse events are assessed consistently across studies.” A citation is given for another review by Ai and colleagues in the Journal of Hematology & Oncology, but this review focused primarily on engineered immune cell engineering and not on toxicity scoring. For a review like this, it would be most helpful to name or even describe the most commonly used toxicity scoring systems. This could be a good opportunity to package information concisely in a Table.

Response 3. Thanks for the comment. The typo regarding citation was corrected. The commonly used methods of toxicity assessment were animal’s survival and weight, blood analysis, general state, and histopathology of organs. This information reflected in table 1.

Response 3. Thanks for the comment. The typo regarding citation was corrected. The commonly used methods of toxicity assessment were animal’s survival and weight, blood analysis, general state, and histopathology of organs. This information reflected in table 1.

Comment 4. Figure 2 is a schematic of potential future directions for optimizing cellular product biosafety. Typically, such schematics represent a concise visual summary of the material discussed in the text. However, there are certain elements in the schematic that are not addressed in the text. For example, under “Strengthening control over transplanted cells,” the following appear only in this schematic and not in the text:

Use optogenetic and chemogenetic tools to control cell behavior

Track distribution and activity with genetic barcodes or liquid biopsy methods

Response 4. Thanks for the comment. We added information regarding optogenetic and chemogenetic tools and genetic barcodes or liquid biopsy methods in section Current limitations and future direction. 

Comment 5. The review is information-dense and text-heavy. It would improve readability immensely to have more sub-sections with their own headings and tables that summarize the content within each section. For example, section 2 identified several key mechanisms of cell therapy toxicity – immunological responses, tumorigenesis, cellular senescence, and administration-related complications. These could be organizational sub-headings within this section. Section 2 also has a paragraph beginning at line 107 that identifies the goals of toxicity studies and key analytical methods. This content would be much more likely to be used by readers and cited if some of it was packaged into a table with key goals, methods, and examples. Similar suggestions could be made in other sections to help this review achieve greater clarity and impact.

Response 5. Thank you so much for your constructive suggestion. We added subsections where it was needed. But we are worried that additional tables will overload the already rather voluminous review.

Comment 6. It is helpful to have summaries of the review's content as the authors provide in Figures 1 and 2. However, the visuals add very little value, in some instances seem irrelevant (e.g., the overlaid histograms, the 96-well plate), and the boxes, arrows, and images occupy space that would be more valuable if it allowed the text to be larger. Given the preponderance of text in Figures 1 and 2, this information would be clearer and easier to navigate if it was displayed in the form of tables with major sub-sections.

Response 6.  Thank you so much for your constructive suggestion. We have reformatted the Figures 1 and 2 for a better understanding of the readers.

Comment 7. The table (I cannot tell the page number since those are missing) lacks a legend and is disproportionately focused compared to the breadth of scope in the rest of the review. The sentence referring to this Table in the text says, “The following biosafety criteria are of critical importance in cell-based research and therapy: oncogenicity, teratogenicity, and tumorigenicity (Table 1).” Yet, the Table conveys information related to the following categories: Toxicity, Tumor Formation, Cell Distribution, and Other. Additionally, unlike the rest of the review and even this section, it is focused exclusively on cell therapies for neurologic diseases. This Table would be much more effective if it aligned more clearly with what is stated in the text, if it had a legend to explain it, and if the rationale for the specific focus on neurological disorders was provided.

Response 7. Thanks for the comment. We added the legend of the table and removed the mention “The following biosafety criteria are of critical importance in cell-based research and therapy: oncogenicity, teratogenicity, and tumorigenicity (Table 1).” We added a mention of the table at lines 613. The table was expanded and cellular products intended for other diseases were added.

Comment 8. The titles for sections 2 and 3 are identical. I suspect one of them is mistaken. This is another opportunity for clearer organizational separation of the dense content of this review.

Response 8. The typo was corrected.

Comment 9. There are examples of cell products being named but not described. For example, on page 2, line 58, there is a reference to rexlemestrocel-L within this sentence, “Another one cell-based drug approved by the FDA is Rexlemestrocel-L which is aimed at therapy of several congenital heart defects.” Given the broad scope of this review, it would be advisable to be specific as to the nature of a product the first time it is mentioned. The sentence given above could be, “Another cell-based drug approved by the FDA is the allogeneic bone marrow-derived mesenchymal precursor cell product Rexlemestrocel-L, which is labeled for the treatment of congenital heart defects.”

Response 9. Thank you for your suggestion. We agree and have changed the sentences to the proposed version.

Comment 10. Overall, this is a relevant, timely, and information-dense review, but it could be much more readable and impactful with improved organization of the text and optimization of the Figures and Table.

Response 10. Thank you for your suggestion. We improved organization of the text and redone the Figures and Table.

Reviewer 2 Report

Comments and Suggestions for Authors

Title: Safety Assessment of Stem Cell–Based Therapies: Current Standards and Advancing Frameworks

Through this manuscript entitled “Safety Assessment of Stem Cell–Based Therapies: Current Standards and Advancing”, Leonov et al. aimed to explore the main biosafety issues in depth, discuss the ongoing challenges within the discipline, and critically examine the tools and methods used to evaluate them. This review offers highly relevant insights in the field, presenting them with clarity and demonstrating a manuscript of remarkable quality. The manuscript is promising; however, some minor revisions are necessary to enhance the quality of this review.

Below are my comments.

Abstract

  • An additional space appears to be present on line 27 of the abstract. The authors should review this section and apply this minor correction if required.

1. Introduction:

  • From lines 40 to 47, the authors explained and listed the two main directions of cell therapy (“Currently, there are two main directions … for the treatment of certain types of cancer”). It would be preferable to divide the sentence into two distinct statements for greater clarity and readability, instead of linking the ideas with a semicolon.
  • A period is missing at the end of the sentence on line 47.
  • From lines 49 to 53, the authors use a long and complex sentence structure. I suggest dividing it into two sentences to make the text clearer and more readable.
  • From lines 49 to 63, the authors repeatedly use the verb “approve.” I suggest replacing it with suitable synonyms throughout the text to make the manuscript less repetitive and improve the flow of the content.
  • In line 58, I suggest replacing “one cell-based drug” with “one-cell-based drug”.
  • In line 59, the authors should replace “therapy” with “the therapy”.
  • In line 60, I suggest replacing “application” with “applications”.
  • From lines 71 to 81, I recommend increasing the connections between sentences to make the text more cohesive and continuous. Additionally, even when similar sentence openings (e.g., “Equally important”) are not consecutive, using synonyms or alternative phrasing can help make the text more varied and readable.
  • From lines 85 to 86, the authors rewrite what they’ve already written at the end of the abstract (lines 28-30). Specifying the aim of the manuscript is important both in the introduction and in the abstract. Nevertheless, to prevent redundancy, the authors are encouraged to express the idea using different wording, which would enhance the overall quality of the manuscript.

2. Assessment of the toxicity of the cellular product:

  • From lines 139 to 180, the terms “assess”, “assessing”, and “assessments” are used multiple times throughout the manuscript. I recommend employing synonyms where appropriate to reduce redundancy. Additionally, I suggest improving the connections between some sentences to avoid presenting a mere list of concepts and to make the paragraph more cohesive and discursive.
  • In line 111, the authors should remove “.

3. Assessment of the toxicity of the cellular product:

  • It is recommended that the authors change the paragraph title, since it duplicates the one used previously.
  • In line 213, after the word “potential” I suggest adding a reference.

4. Assessment of Immunogenicity in Cell Therapy

  • The paragraph is clear, well-written, and accurately conveys the content. Nevertheless, between lines 327 and 347, multiple sentences (some consecutive) start with “This study showed.” As previously noted for other sections, it is advisable to avoid this repetition by varying the phrasing. Moreover, when introducing a scientific work for the first time, generic expressions should be replaced with specific references, preferably using the first author’s name followed by et al.
  • PCR is cited as the methodology on line 343. However, subsequent references to cycle threshold (Ct) values imply that quantitative information is being used. Thus, referring simply to PCR is not precise, as the technique employed is qRT-PCR. I recommend clarifying this point and providing a clearer explanation of why the authors interpret the different Ct values (32 vs. 24) as indicative of immunotolerance. Including an explanatory sentence would make the discussion more complete.
  • Reference 7 (line 347) appears to be incorrect, as its DOI is the same as that of reference 53, and the title does not seem accurate. The authors are advised to carefully revise this reference and verify the titles and DOIs of all references, ensuring there are no duplications or misnumbered citations.
  • References are missing in the final paragraph of this section. They should be inserted in line 365 after “transplantation”, line 367 after “transplantation”, and at the end of line 372 after “NKs”.

Table:

  • The table presents multiple reference inaccuracies, as the numbers do not match the corresponding references. The authors are advised to revise and correct the numbering for references 34, 63, 64, 65, and 96.

7. Current limitations and future direction:

  • From line 593 onwards, multiple sentences begin with “These” or “Furthermore,” resulting in some repetition. Consistent with recommendations made for other sections, I suggest employing synonyms or alternative sentence structures to enhance readability and flow. Furthermore, lines 595–596 use “regulatory” followed immediately by “regulators”; the authors are advised to revise these terms to reduce redundancy.

Apart from these minor revisions, the manuscript is of excellent quality and presents the information in an accurate and comprehensive manner. Therefore, in my opinion, it deserves to be published.

Author Response

We are grateful to the reviewer for evaluating our work! The authors very much appreciated the constructive comments on this manuscript by the reviewer. The comments have been very thorough and useful in improving the manuscript.

Comment 1. From lines 40 to 47, the authors explained and listed the two main directions of cell therapy (“Currently, there are two main directions … for the treatment of certain types of cancer”). It would be preferable to divide the sentence into two distinct statements for greater clarity and readability, instead of linking the ideas with a semicolon.

Response 1: The sentence has been divided into two distinct statements for improved clarity and readability. The remark has been corrected.

Comment 2. A period is missing at the end of the sentence on line 47.

Response 2: The remark has been corrected.

Comment 3. From lines 49 to 53, the authors use a long and complex sentence structure. I suggest dividing it into two sentences to make the text clearer and more readable.

Response 3: The remark has been corrected.

Comment 4. From lines 49 to 63, the authors repeatedly use the verb “approve.” I suggest replacing it with suitable synonyms throughout the text to make the manuscript less repetitive and improve the flow of the content.

Response 4: The remark has been corrected.

Comment 5. In line 58, I suggest replacing “one cell-based drug” with “one-cell-based drug”.

Response 5: The remark has been corrected.

Comment 6. In line 59, the authors should replace “therapy” with “the therapy”.

Response 6: The remark has been corrected.

Comment 7.n line 60, I suggest replacing “application” with “applications”.

Response 7: The remark has been corrected.

Comment 8. From lines 71 to 81, I recommend increasing the connections between sentences to make the text more cohesive and continuous. Additionally, even when similar sentence openings (e.g., “Equally important”) are not consecutive, using synonyms or alternative phrasing can help make the text more varied and readable.

Response 8: Connections between sentences have been improved. The remark has been corrected.

Comment 9.  From lines 85 to 86, the authors rewrite what they’ve already written at the end of the abstract (lines 28-30). Specifying the aim of the manuscript is important both in the introduction and in the abstract. Nevertheless, to prevent redundancy, the authors are encouraged to express the idea using different wording, which would enhance the overall quality of the manuscript.

Response 9: The abstract was rewritten. 

Comment 10. From lines 139 to 180, the terms “assess”, “assessing”, and “assessments” are used multiple times throughout the manuscript. I recommend employing synonyms where appropriate to reduce redundancy. Additionally, I suggest improving the connections between some sentences to avoid presenting a mere list of concepts and to make the paragraph more cohesive and discursive.

Response 10: The remark has been corrected.

Comment 11. In line 111, the authors should remove “.

Response 11: The remark has been corrected.

Comment 12. It is recommended that the authors change the paragraph title, since it duplicates the one used previously.

Response 12: The authors agree with this comment. We have made appropriate changes to the manuscript.

Comment 13. In line 213, after the word “potential” I suggest adding a reference.

Response 13: We have made appropriate changes to the manuscript.

Comment 14. The paragraph is clear, well-written, and accurately conveys the content. Nevertheless, between lines 327 and 347, multiple sentences (some consecutive) start with “This study showed.” As previously noted for other sections, it is advisable to avoid this repetition by varying the phrasing. Moreover, when introducing a scientific work for the first time, generic expressions should be replaced with specific references, preferably using the first author’s name followed by et al.

Response 14: Corrected.  We have made appropriate changes to the manuscript.

Comment 15. PCR is cited as the methodology on line 343. However, subsequent references to cycle threshold (Ct) values imply that quantitative information is being used. Thus, referring simply to PCR is not precise, as the technique employed is qRT-PCR. I recommend clarifying this point and providing a clearer explanation of why the authors interpret the different Ct values (32 vs. 24) as indicative of immunotolerance. Including an explanatory sentence would make the discussion more complete.

Response 15: The wording of the sentence has been corrected.

Comment 16. Reference 7 (line 347) appears to be incorrect, as its DOI is the same as that of reference 53, and the title does not seem accurate. The authors are advised to carefully revise this reference and verify the titles and DOIs of all references, ensuring there are no duplications or misnumbered citations.

Response 16: The duplicate reference has been removed. We have made appropriate changes to the manuscript.

Comment 17. References are missing in the final paragraph of this section. They should be inserted in line 365 after “transplantation”, line 367 after “transplantation”, and at the end of line 372 after “NKs”.

Response 17: Corrected. The relevant references have been added to the text of the manuscript.

Comment 17. The table presents multiple reference inaccuracies, as the numbers do not match the corresponding references. The authors are advised to revise and correct the numbering for references 34, 63, 64, 65, and 96.

Response 17: Corrected.  We have made appropriate changes to the manuscript.

Comment 18. From line 593 onwards, multiple sentences begin with “These” or “Furthermore,” resulting in some repetition. Consistent with recommendations made for other sections, I suggest employing synonyms or alternative sentence structures to enhance readability and flow. 

Response 18: Corrected.  We have made appropriate changes to the manuscript.

Comment 19. Furthermore, lines 595–596 use “regulatory” followed immediately by “regulators”; the authors are advised to revise these terms to reduce redundancy.

Response 19: Corrected.  We have made appropriate changes to the manuscript.

Reviewer 3 Report

Comments and Suggestions for Authors

The review clearly organizes the key biosafety pillars for cell-therapy evaluation. Please address the section-specific comments below to improve structure, accuracy, and practical utility.

1. In abstract, state explicitly what this review adds beyond existing biosafety overviews (e.g., a side-by-side assay toolkit or a regulator comparison). Replace generalities with specifics: list the four pillars the authors actually synthesize (toxicity, oncogenicity/tumorigenicity/teratogenicity, immunogenicity, biodistribution) and one concrete take-home per pillar.

2. In introduction part, the Tecelra (afamitresgene autoleucel) and Amtagvi (lifileucel) examples are appropriate; keep them with exact label language and dates. However, “Rexlemestrocel-L … approved by the FDA” appears inconsistent with the trial-oriented reference now cited; please confirm the current regulatory status via primary FDA sources or revise to “in trials/under review,” to avoid a factual error.

3. Section 2: Assessment of toxicity. Separate ‘general tox’ from ‘biosafety pharmacology’. The authors discuss both but they blur together. I suggest two clear subsections: (a) General systemic & local toxicity (clinical signs, clinical chemistry, histopathology, dose setting, route-relevance), and (b) Safety pharmacology (CV/respiratory/neurologic panels and triggers to expand). If feasible, add a short table listing minimal data sets for local vs systemic administration, including units (cells/kg), viability %, and acceptance criteria linked to release tests, this will improve translational utility

4. There are two sections titled “Assessment of the toxicity of the cellular product”; the second actually treats oncogenicity/tumorigenicity/teratogenicity. Please retitle it to “3. Oncogenicity, Tumorigenicity, and Teratogenicity” and fix downstream numbering.

5. Section 4: Immunogenicity: the author cover innate assays (NK degranulation/CD107a, LDH) and MLR; consider adding ELISpot (IFN-γ), donor-specific anti-HLA antibody monitoring, and complement activation assays (C3a/C5a) especially for repeated dosing or matrix-containing products. For hypoimmunogenic (“universal”) cells, add a balancing note on NK escape strategies (e.g., HLA-E/HLA-G overexpression) and the need for in vivo confirmation under inflammatory cytokine exposure (IFN-γ up-regulates HLA).

6. Standardize terms (e.g., “biodistribution” not “bio-distribution”), correct typos (“transplantate” → “transplant”). Check thoroughly the entire manuscript.

Author Response

Dear Reviewer,

We are grateful for your time and effort spent reviewing our manuscript. We highly appreciate your constructive feedback and thoughtful comments. We revised the manuscript according to your comments. All changes were highlighted in yellow color.

Comment 1. In abstract, state explicitly what this review adds beyond existing biosafety overviews (e.g., a side-by-side assay toolkit or a regulator comparison). Replace generalities with specifics: list the four pillars the authors actually synthesize (toxicity, oncogenicity/tumorigenicity/teratogenicity, immunogenicity, biodistribution) and one concrete take-home per pillar.

Response 1. Thank you for your suggestion. We rewrote abstract section according your comment.

Comment 2. In introduction part, the Tecelra (afamitresgene autoleucel) and Amtagvi (lifileucel) examples are appropriate; keep them with exact label language and dates. However, “Rexlemestrocel-L … approved by the FDA” appears inconsistent with the trial-oriented reference now cited; please confirm the current regulatory status via primary FDA sources or revise to “in trials/under review,” to avoid a factual error.

Response 2. Thank you for your comment. We added data about Tecelra (afamitresgene autoleucel) and Amtagvi (lifileucel), and corrected the inaccuracy. 

Comment 3. Section 2: Assessment of toxicity. Separate ‘general tox’ from ‘biosafety pharmacology’. The authors discuss both but they blur together. I suggest two clear subsections: (a) General systemic & local toxicity (clinical signs, clinical chemistry, histopathology, dose setting, route-relevance), and (b) Safety pharmacology (CV/respiratory/neurologic panels and triggers to expand). If feasible, add a short table listing minimal data sets for local vs systemic administration, including units (cells/kg), viability %, and acceptance criteria linked to release tests, this will improve translational utility

Response 3. Thank you for your suggestion. We separated the section ‘Assessment of toxicity’ according your suggestion and added new information in subsection ‘biosafety pharmacology’. But we are worried that additional tables will overload the already rather voluminous review, and information regarding administration, including units (cells/kg), viability added in Table 1.

Comment 4. There are two sections titled “Assessment of the toxicity of the cellular product”; the second actually treats oncogenicity/tumorigenicity/teratogenicity. Please retitle it to “3. Oncogenicity, Tumorigenicity, and Teratogenicity” and fix downstream numbering.

Response 4.  Thank you for your comment. This typo was corrected.

Comment 5. Section 4: Immunogenicity: the author cover innate assays (NK degranulation/CD107a, LDH) and MLR; consider adding ELISpot (IFN-γ), donor-specific anti-HLA antibody monitoring, and complement activation assays (C3a/C5a) especially for repeated dosing or matrix-containing products. For hypoimmunogenic (“universal”) cells, add a balancing note on NK escape strategies (e.g., HLA-E/HLA-G overexpression) and the need for in vivo confirmation under inflammatory cytokine exposure (IFN-γ up-regulates HLA).

Response 5. Thank you for your suggestion. We added information regarding ELISpot (IFN-γ), donor-specific anti-HLA antibody monitoring, and complement activation assays (C3a/C5a) lines 391-399, 416-421, 437-446. Also we added information about NK escape strategies (HLA-E/HLA-G overexpression) for hypoimmunogenic (“universal”) cells, and influence of exposure inflammatory on HLA modulation lines 408-415, 468-471.

Comment 6. Standardize terms (e.g., “biodistribution” not “bio-distribution”), correct typos (“transplantate” → “transplant”). Check thoroughly the entire manuscript.

Response 6. Thank you for your comment. This typo was corrected.